# Functional double dissociation within the entorhinal cortex for visual scene-dependent choice behavior

**Seung-Woo Yoo, Inah Lee***

Department of Brain and Cognitive Sciences, Seoul National University, Seoul, Korea

**Abstract** How visual scene memory is processed differentially by the upstream structures of the hippocampus is largely unknown. We sought to dissociate functionally the lateral and medial subdivisions of the entorhinal cortex (LEC and MEC, respectively) in visual scene-dependent tasks by temporarily inactivating the LEC and MEC in the same rat. When the rat made spatial choices in a T-maze using visual scenes displayed on LCD screens, the inactivation of the MEC but not the LEC produced severe deficits in performance. However, when the task required the animal to push a jar or to dig in the sand in the jar using the same scene stimuli, the LEC but not the MEC became important. Our findings suggest that the entorhinal cortex is critical for scene-dependent mnemonic behavior, and the response modality may interact with a sensory modality to determine the involvement of the LEC and MEC in scene-based memory tasks.

## Introduction

The visual scene appears to be a strong sensory attribute that functionally recruits the hippocampal system based on human studies (*Epstein and Kanwisher, 1998*; *Hartley et al., 2007*; *Hassabis et al., 2007*; *Staresina et al., 2011*; *Zeidman et al., 2015*) and nonhuman primate studies (*Gaffan and Harrison, 1989*; *Wirth et al., 2003*). Memory for the visual scene may function as the linchpin for explaining the involvement of the hippocampus in various cognitive tasks requiring spatial navigation, episodic memory, and imagining future events [see *Maguire and Mullally (2013)* for review]. In our earlier studies, we confirmed that this involvement was also true in rats by showing that rats were impaired in making behavioral choices using background visual scenes when the dorsal hippocampus was inactivated by a GABA-A receptor agonist, muscimol (MUS) (*Kim et al., 2012*; *Delcasso et al., 2014*; *Lee et al., 2014*). In addition, hippocampal neurons showed significant modulation in the firing rate in a visual scene-dependent manner (*Delcasso et al., 2014*).

Despite the significant involvement of the hippocampus in visual scene-dependent memory, the functional significance of its upstream structures, that is, the medial entorhinal cortex (MEC) and lateral entorhinal cortex (LEC), is unclear. Because the entorhinal cortex (EC) provides major inputs to the hippocampus (*Lavenex and Amaral, 2000*; *Witter et al., 2000*; *Knierim et al., 2006*; *Kerr et al., 2007*), the EC is expected to play key roles in scene memory. However, the visual scene has not been considered a key factor in most theories for understanding information processing in the hippocampal system (but see *Knierim et al., 2014*). Specifically, a majority of recent studies have been guided by a theory that proposed differential functions of the LEC and MEC for nonspatial and spatial memory, respectively (*Hargreaves et al., 2005*; *Knierim et al., 2006*; *Henriksen et al., 2010*; *Ito and Schuman, 2012*; *Tsao et al., 2013*; *Van Cauter et al., 2013*). According to the theory, the LEC receives more projections from the perirhinal cortex (PER), an area that is well known for its role in object recognition memory, and the LEC projects to the

*For correspondence: lee.inah@gmail.com

**Competing interests:** The authors declare that no competing interests exist.

hippocampus and subiculum. The MEC receives most of the projections from the postrhinal cortex (POR), the rodent homolog of the parahippocampal cortex in primates and humans, and projects to the hippocampus and subiculum. It has been hypothesized that LEC-dependent nonspatial memory and MEC-dependent spatial memory are combined as an integrative memory representation within the hippocampus. Physiological studies that have reported differential functions between the LEC and MEC in recent years seemingly support this theoretical framework (*Hargreaves et al., 2005*; *Henriksen et al., 2010*; *Ito and Schuman, 2012*; *Tsao et al., 2013*). For example, the presence of grid cells in the MEC is believed to provide key evidence in support of the spatial functions of the MEC (*Hafting et al., 2005*), and the cells in the LEC fire in a nonspatial manner (*Hargreaves et al., 2005*).

According to the dominant theory described above, it is implied that the visual scene information is processed by the spatial information processing stream involving the POR and MEC (*Knierim et al., 2014*). That is, the POR is characterized as receiving 'visuospatial' sensory information, and the visual scene may belong to this information category (*Burwell and Amaral, 1998*). Because nonspatial information is often characterized based on individual sensory attributes (e.g., olfactory, tactile, and auditory information) from the unimodal and polymodal sensory areas (*Burwell and Amaral, 1998*) that may be associated more with individual objects than scenes, the existing theory may predict that visual scene-dependent memory is uniquely processed by the spatial information-processing stream involving the POR and MEC, but not the PER and LEC. Although this idea might be supported indirectly by behavioral studies testing animals with spatial or contextual memory tasks (*Majchrzak et al., 2006*; *Hunsaker et al., 2013*; *Van Cauter et al., 2013*; *Morrissey and Takehara-Nishiuchi, 2014*), visual scene has hardly been systematically controlled in prior studies that examined the functions of the LEC and MEC.

Another important point that has been overlooked is that sensory and motor components appear to be mixed when existing theories characterize the spatial information stream in particular. For example, if one makes lesions in the POR or MEC and obtains performance deficits in a behavioral task (usually a spatial navigation task), it is difficult to know whether the animal made errors because visuospatial sensory information processing was impaired or because the lesions damaged the response component associated with making spatial choices. The dominant theory seemingly divides the spatial and nonspatial information-processing streams based on the characteristics of sensory information (*Burwell and Amaral, 1998*; *Witter et al., 2000*; *Eichenbaum et al., 2012*), but this division may be an oversimplification of how the brain works. Examining this issue requires testing animals in various memory tasks that share the same sensory stimuli but require different motor responses (e.g., spatial versus nonspatial responses).

In the current study, we examined whether the LEC and MEC were differentially involved in visual scene memory by directly comparing the results of reversibly inactivating the LEC and MEC separately within the same rats. We used a scene-dependent spatial choice (SSC) task and a scene-dependent nonspatial choice (SNSC) task because, based on our prior studies (*Kim et al., 2012*; *Delcasso et al., 2014*; *Lee et al., 2014*), both tasks are hippocampal-dependent but they require spatial and nonspatial responses, respectively. If visual scene memory is exclusively processed in the MEC, inactivating the MEC, but not the LEC, should result in severe deficits in performance in both SSC and SNSC tasks. Here, for the first time to our knowledge, we report a functional double dissociation between the LEC and MEC in visual scene memory. Our results suggest that visual scene information is not uniquely processed in the MEC, and the type of motor response also seems to be important in determining the involvement of entorhinal subdivisions in scene memory tasks.

## Results

### Double dissociation between the LEC and MEC in visual scene-based choice behavior

We tested rats (n = 8) implanted with bilateral cannulae targeting both the LEC and MEC simultaneously (*Figure 1A* to *Figure 1C*) in a scene-based spatial choice (SSC) task in which the rats were required to choose either the left or right arm in a T-maze using the surrounding visual scene displayed in an array of LCD panels around the maze (*Kim et al., 2012*; *Delcasso et al., 2014*) (*Figure 2A*; *Video 1*). The rats performed at above 90% correct when they were retested after

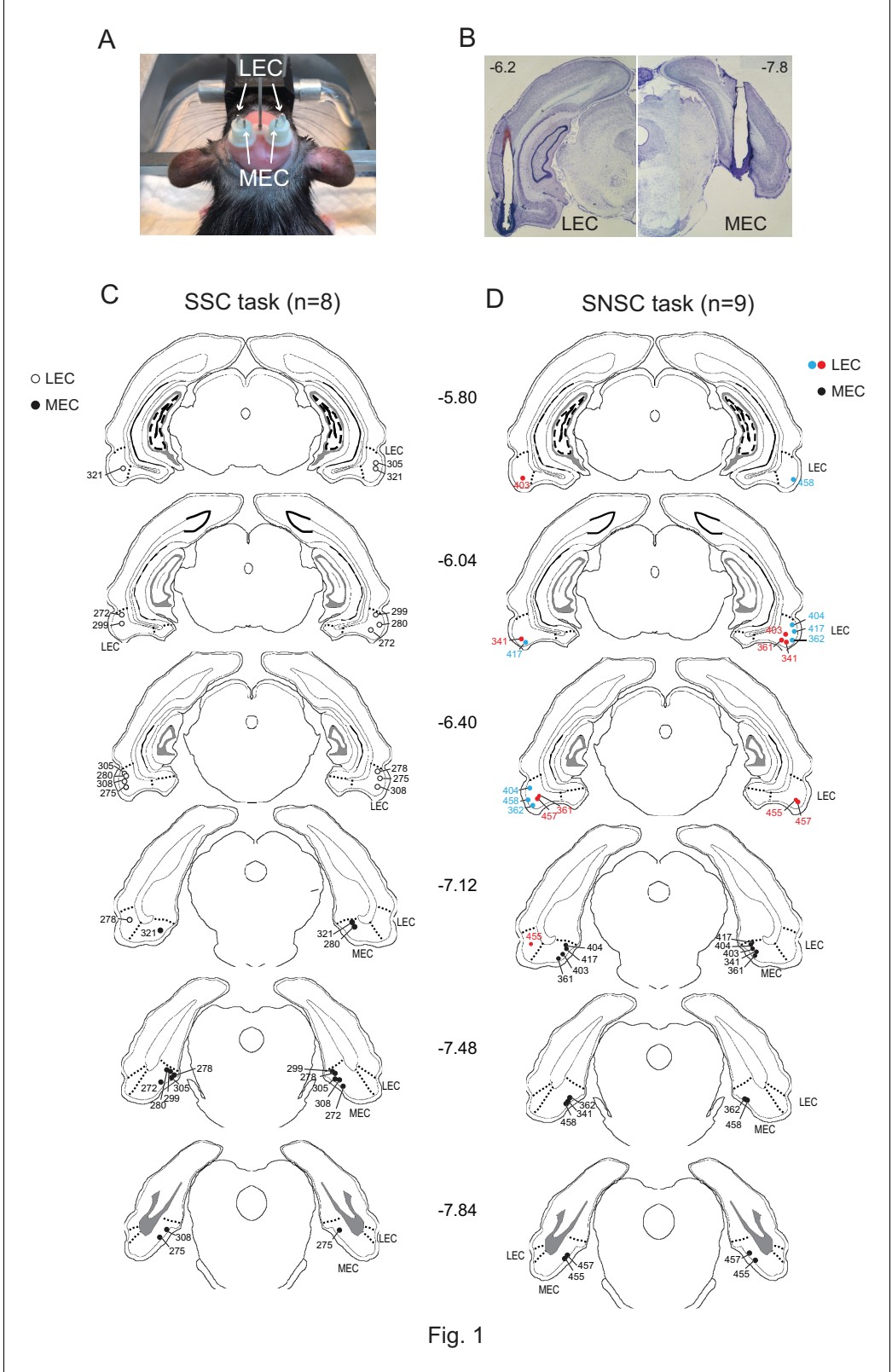

Fig. 1

**Figure 1.** Bilateral cannula implantations in the LEC and MEC within the same rat and histological results. (**A**) Custom-made 3D-printed cannula complex. (**B**) Representative photomicrographs showing the cannula tracks in the LEC (upper left) and MEC (upper right) from the same rat. The numbers denote the relative positions from bregma (mm). (**C**) Locations of cannula tips of all rats used in the SSC task (n = 8). (**D**) Locations of cannula

*Figure 1 continued on next page*

*Figure 1 continued*

tips of all rats used in the SNSC task (n = 9). For LEC cannula locations, red circles indicate medially placed cannula positions in the LEC and blue circles indicate laterally placed cannula positions. The numbers besides the sections denote relative positions of the sections from bregma (mm).

surgery with artificial cerebrospinal fluid (aCSF) injected into the EC (EC-aCSF) (*Figure 2B*; *Figure 2—source data 1*). When MUS was injected into the LEC (LEC-MUS) on the next day, performance did not significantly change. However, MUS injections into the MEC (MEC-MUS) on the following day significantly impaired rats' performance. A repeated-measures ANOVA showed a significant effect of the drug ($F_{(2,14)}$ = 19.3, p<0.0001). A Bonferroni-Dunn *post-hoc* test revealed

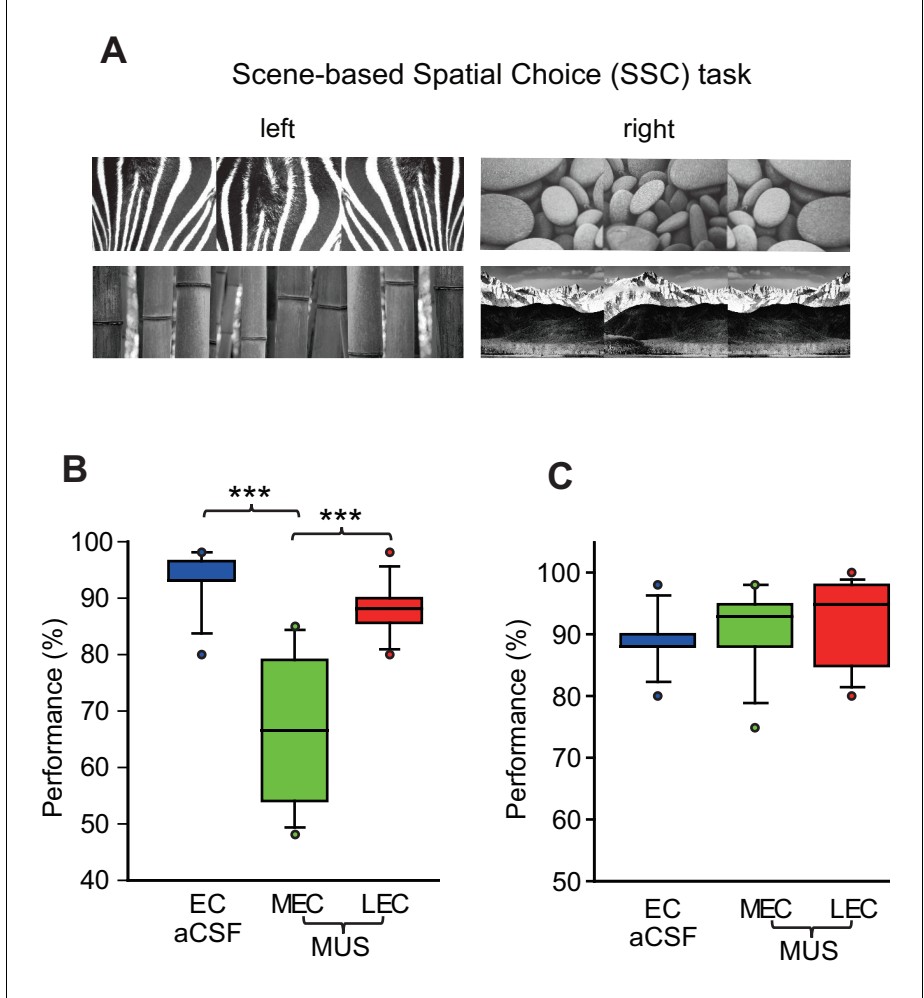

**Figure 2.** Visual scene-based and tactile cue-based spatial choice tasks. (**A**) Visual scene stimuli that were used in the SSC task. The two scenes were associated with the left turn, and the other two scenes were associated with the right turn. (**B**) Post-surgical performance in the SSC task. The MEC-MUS group exhibited significant performance deficits compared to the EC-aCSF and LEC-MUS groups. (**C**) Performance in the tactile cue-based choice task in the T-maze.

The following source data is available for figure 2:

**Source data 1.** (SSC).
**Source data 2.** (Tactile).

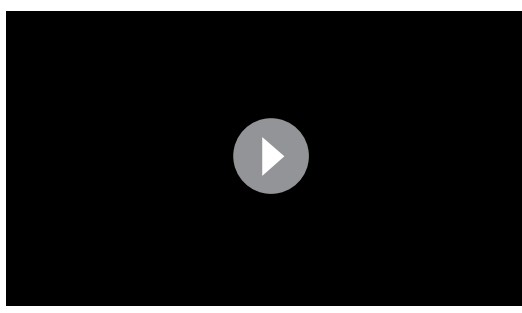

**Video 1.** SSC task. Normal performance of rats in the SSC task. Sample trials, each associated with zebra stripes, bamboos, mountains, or pebbles scene are shown in the video.

significant differences in performance between animals in the vehicle group and the MEC-MUS group (p<0.0001), as well as between the LEC-MUS and MEC-MUS groups (p<0.001). There were no significant differences in choice latency among the drug conditions ($F_{(2,14)}$ = 0.2, p=0.8; repeated-measures ANOVA), suggesting that the impairment in performance in the MEC-MUS condition was not attributable to generic sensory-motor deficits. Furthermore, the MEC-MUS did not disrupt the rat's capability of associating spatial responses with any type of sensory stimulus because the same rats were able to make spatial choices (i.e., left and right turns) normally under the MUS conditions when tactile cues (wire mesh or sandpaper) were used as cues instead of visual scenes ($F_{(2,18)}$ = 1.39, p=0.27; repeated-measures ANOVA) (*Figure 2C*; *Figure 2—source data 2*).

To test whether the MEC-dependent performance deficits in the SSC task could be observed also when a visual scene must be associated with a nonspatial response, we tested a different group of rats (n = 9) in the SNSC task in which nonspatial choice behavior (digging or pushing) was required to manipulate a sand-filled jar in association with visual scenes (*Lee and Shin, 2012*; *Lee et al., 2014*) (*Figure 3A*; *Video 2*). The animals performed at approximately 90% correct when they were retested after surgery with aCSF injected into the EC. To our surprise, MUS injections into the LEC (LEC-MUS), however, significantly disrupted performance in the same rats, whereas MUS did not produce a significant impairment in performance in the MEC-MUS condition (*Figure 3B*; *Figure 3—source data 1*). A repeated-measures ANOVA showed a significant effect of drug on performance ($F_{(2,16)}$ = 10.1, p=0.001). A *post-hoc* test (Bonferroni-Dunn) showed significant differences between the EC-aCSF and LEC groups (p<0.001) and between the LEC-MUS and MEC-MUS groups (p=0.01). No significant effect of drug was found for choice latency ($F_{(2,16)}$ = 0.29, p=0.74).

Despite the significant impairment in performance with MUS injections in the LEC, some rats seemed less affected by LEC-MUS, which was noticeable in the large variance in the performance data for the LEC-MUS condition compared to the other drug conditions (*Figure 3B*). We examined whether there was a relationship between the locations of the cannula tip positions in the LEC and performance levels in individual rats. If one of the cannula was only placed laterally, the animal's cannula position was still labeled lateral (e.g., rat 417 in *Figure 1D*). Interestingly, it turns out that the rats with their LEC cannulae implanted more laterally along the superficial layers (marked in blue circles in *Figure 1D*) were not as severely impaired as the animals with their LEC cannulae implanted more medially along the deeper layers (marked in red circles in *Figure 1D*) when performance of the two subgroups under MUS were directly compared ($t_{(7)}$ = −4.35, p<0.01) (*Figure 3C*). The two subgroups were not significantly different from each other in the aCSF condition ($t_{(7)}$ = 0.3, p=0.77) (*Figure 3C*).

The functional double dissociation between the LEC and MEC can be more clearly visible when the amount of performance deficit (measured by subtracting performance under MUS from performance under aCSF) was plotted for all MUS conditions for the LEC and MEC (*Figure 4*). An ANOVA was performed with the task as a between-group factor and the drug condition as a within-group factor, revealing a highly significant interaction between the task and drug conditions ($F_{(1,15)}$ = 34.75, p<0.0001). *Post-hoc* tests (Bonferroni-corrected t-tests) showed that the LEC-MUS condition produced significantly bigger deficits compared to the control condition in the SNSC task (p<0.05), but not in the SSC task (p>0.5), whereas the reverse was true in the SSC task (p=0.001 with MEC-MUS).

These results collectively show that the LEC and MEC were both involved in visual scene-based memory tasks, although different task demands recruited the two regions differentially. That is, the LEC was important when rats made nonspatial choices (digging *versus* pushing) toward the same

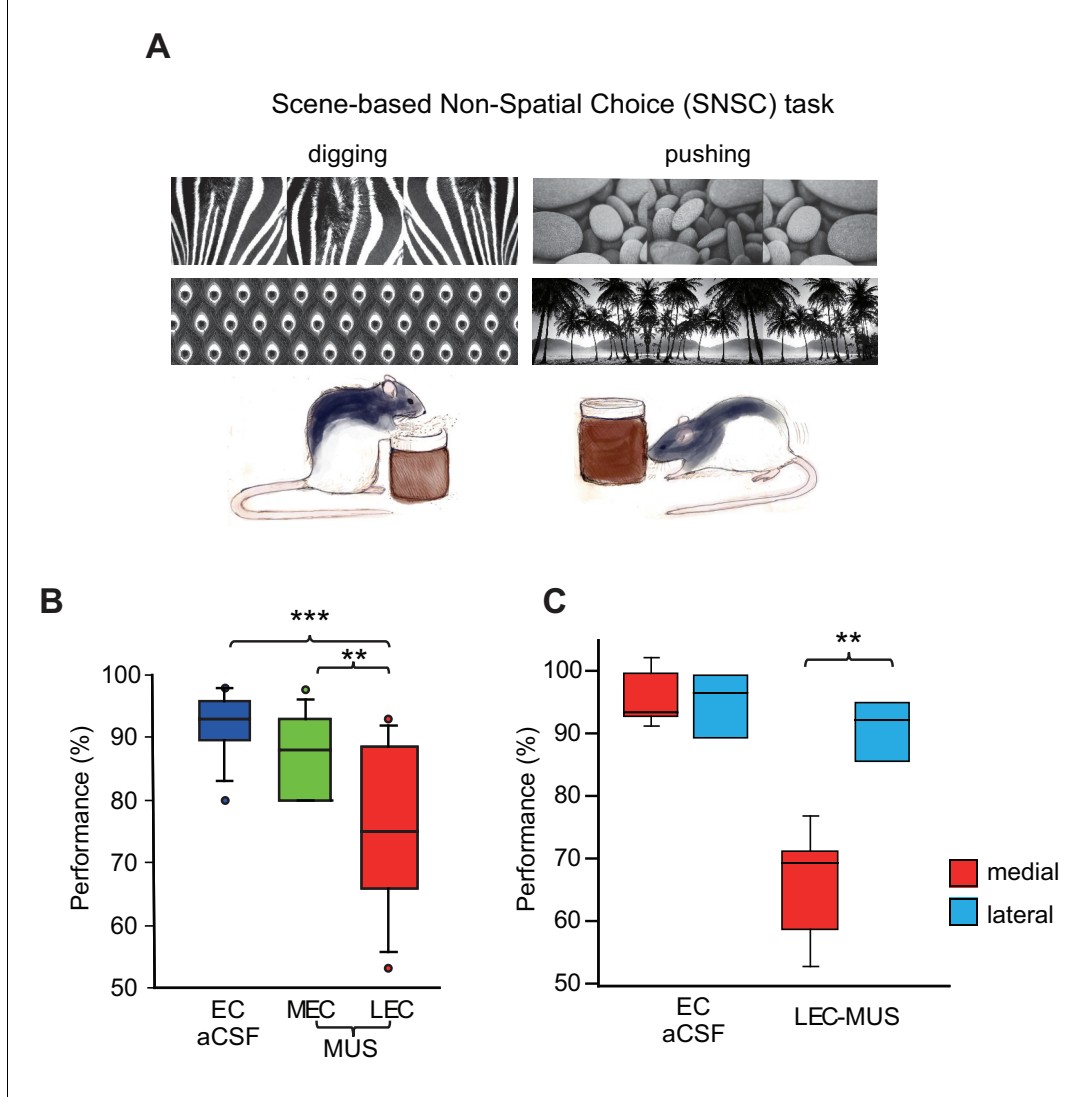

**Figure 3.** Visual scene-dependent nonspatial memory task and behavioral performance. (**A**) Visual scene stimuli and their associated choice responses in the SNSC task. (**B**) Post-surgical performance in the SNSC task. The LEC-MUS condition induced significant performance deficits compared to the EC-aCSF condition. **p<0.01, ***p<0.0001. (**C**) Rats with the LEC cannulae implanted more medially showed bigger deficits in performance than those with more lateral LEC cannulae, whereas those rats were not different from each other when aCSF was injected.

The following source data is available for figure 3:

**Source data 1.** (SNSC).

object (sand-filled jar) using visual scenes in the background, but not for spatial choices (left turn *versus* right turn). Conversely, the MEC was more involved in the SSC task but not in the SNSC task.

## No involvement of the LEC and MEC when an object alone cued choice behavior

It is important to note that compared to the SSC task, the SNSC task required the animal to interact with the object (sand-filled jar). Because several studies have described the roles of the LEC in item-based memory (*Deshmukh and Knierim, 2011*; *Hunsaker et al., 2013*; *Keene et al., 2016*), the performance deficits of the animals in the LEC-MUS group during the SNSC task might be attributable to the involvement of the object, but not necessarily to the involvement of visual scene-based memory. Therefore, to test whether the object factor alone (without scenes) could result in performance

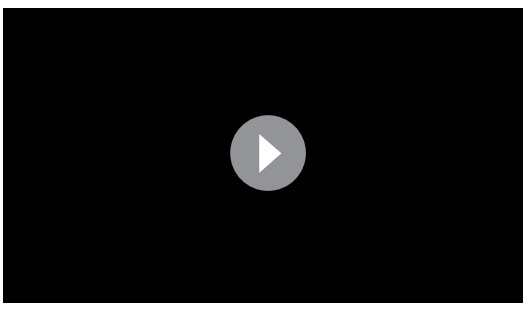

**Video 2.** SNSC task. Performance of rats when injected with aCSF in the EC versus MUS in the LEC in the SNSC task. Scenes are not shown in the video.

deficit, we tested subsets of rats from both tasks (n = 6 from the SSC task and n = 3 from the SNSC task) in an object-based nonspatial choice task (*Figure 5A*). In this task, rats were required to push or dig the sand-filled jar after sampling an object cue attached to the jar. The task thus required the rats to use the same response type (i.e., digging and pushing responses) used in the SNSC task, but instead of using background visual scenes, rats made choices using only objects as cues.

When injected with aCSF in the EC after being trained to criterion in the object-based choice task, rats produced correct responses on more than 90% of the trials (*Figure 5B*; *Figure 5— source data 1*), and the injection of MUS in neither the LEC nor MEC produced performance deficits compared to the aCSF condition ($F_{(2,16)}$ = 1, p=0.37; repeated-measures ANOVA). These results clearly demonstrate that neither subdivision of the EC was necessary when rats used a single object to make a digging versus pushing response to the jar. The performance level under aCSF (mean = 92.9%) in the object memory task was equivalent to the performance levels of the control conditions in the SSC (mean = 92.8%) and SNSC (mean = 91.7%) tasks ($F_{(2,23)}$ = 0.13, p=0.88; ANOVA), suggesting that the null effect of MUS in the object memory task was not attributable to the ceiling effect for performance in the control condition. The results also demonstrate that the digging and pushing behaviors themselves were not necessarily affected by MUS injections into the LEC or MEC. Our findings thus suggest that the performance impairment observed in animals that received the MUS injection into the LEC in the SNSC task may not be solely based on the presence of the object variable compared to the SSC task.

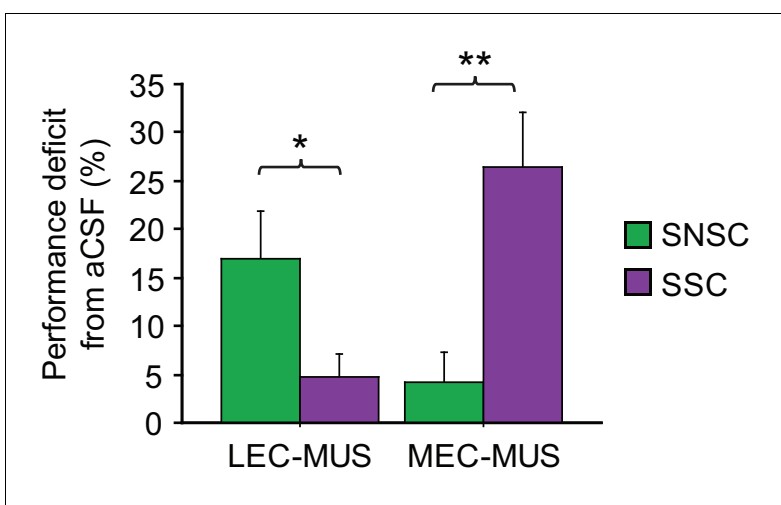

**Figure 4.** Double dissociation between the LEC and MEC in the SSC and SNSC tasks. Ordinate represents the amount of performance difference between aCSF and MUS conditions. *p<0.05, **p<0.001.

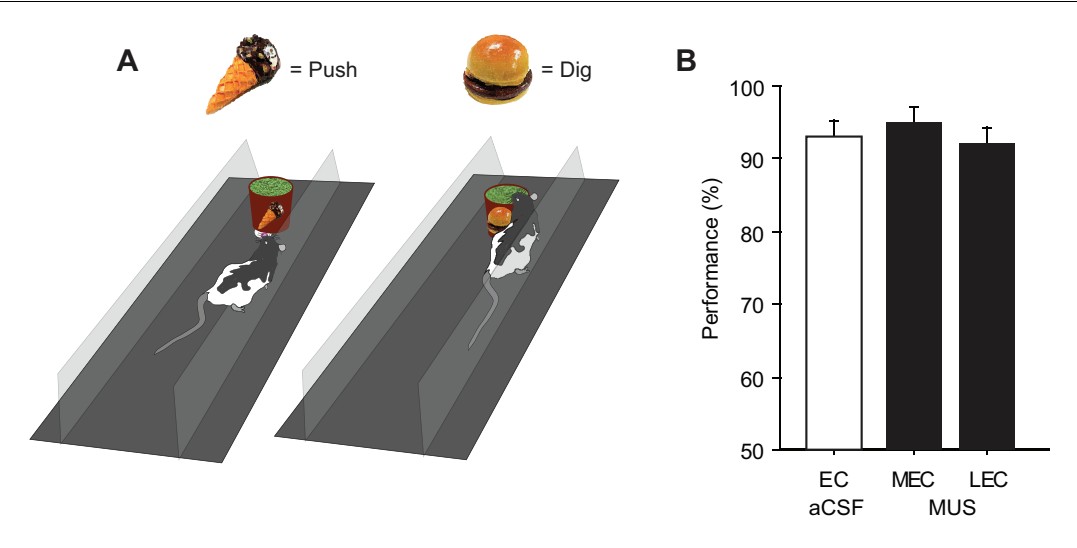

**Figure 5.** Illustration of the object-based choice task and post-surgical performance. (**A**) In the task, the rats were trained to respond either by digging or pushing the jar based on the magnetic object cue attached to the jar. (**B**) Post-surgical performance on the task. No significant differences were found between animals in either drug condition.

The following source data is available for figure 5:

**Source data 1.** (Object).

## Discussion

### Visual scene memory-dependent spatial information processing in the MEC

In the current study, the inactivation of the MEC severely disrupted performance on the SSC task, but not on the SNSC task, whereas the inactivation of the LEC resulted in the opposite pattern of results. However, the MEC-MUS rats were normal in making spatial choices in response to tactile cues. These findings strongly suggest that the MEC is critically involved in using background *visual scenes* for making spatial responses (e.g., left or right turn response). It is possible that the rodent POR is uniquely involved in scene information processing, as in the parahippocampal cortex in humans (*Epstein and Kanwisher, 1998*), and the MEC, a downstream structure of the POR, is also heavily involved in processing scene representations. However, because MEC-MUS rats were not impaired on the SNSC task, the MEC may not be responsible for simply processing visual scenes in any type of behavior. Instead, our SSC task required the rat to associate external visual scene information with an egocentric spatial response (left or right turn). Considering that the MEC is connected to a number of areas in the brain that process idiothetic spatial information (e.g., the parasubiculum, retrosplenial cortex, etc.) (*McNaughton et al., 2006*; *Knierim et al., 2014*), the performance deficits observed in the current study with MEC-MUS in the SSC task might be related to the roles of the MEC in associating egocentric spatial information with visual scenes. Our experimental data from the tactile memory task suggest that such associative function of the MEC is uniquely related to visual scenes because the MEC was not involved in associating tactile cues with egocentric responses. By contrast, the LEC is not connected to those egocentric spatial areas, which may explain why the LEC was not significantly involved in the SSC task. Our results suggest that the hippocampus relies heavily on MEC inputs in order for the rat to perform the SSC task normally.

### Visual scene-based disambiguation of object-associated responses in the LEC

The results from the previous studies showed that the effects of the LEC lesions became more critical when objects were recognized in association with their positions in the testing arena

(*Hunsaker et al., 2013*; *Van Cauter et al., 2013*; *Wilson et al., 2013a*, *2013b*; *Rodo et al., 2017*). The significant decrease in performance in the SNSC task after MUS injections in the LEC (but not in the MEC) may support those previous findings because the rat should make discrete choices when facing the common (thus ambiguous) object in the current task, and only the visual scene information disambiguated which response was appropriate. The presence of the object by itself may not explain the MUS-induced deficits in the LEC because, as shown in prior studies (*Van Cauter et al., 2013*; *Wilson et al., 2013a*, *2013b*), using an object alone as a cue did not produce a significant impairment in performance in the LEC-MUS group in our object memory task. It is possible that the LEC becomes critical when object-associated ambiguity should be reduced using scene or contextual information. In a spontaneous object exploration paradigm, object ambiguity may increase as an animal experiences more objects in the same environment (*Hunsaker et al., 2013*; *Van Cauter et al., 2013*). In our study, the object-associated ambiguity may stem from the task demand that two different behavioral responses should be conditionally emitted to the same object.

In our study, the rats with LEC cannulae implanted more medially along the deeper layers showed more severe deficits than the rats with more laterally implanted cannulae in the LEC. It is possible that the inactivating agent injected through the laterally implanted cannulae near the cortical surface may have been leaked partially to the cortical surface instead of being fully injected into the LEC. It is also possible that the deeper layers of the LEC play more important roles in the SNSC task than the superficial layers of the LEC, possibly because the deeper layers of the LEC receive the inputs from the distal CA1 and thus may have access to both object and place information, which may in turn play critical roles in contextually responding to an object in different contexts.

Based on our surgical coordinates, the LEC cannulae inevitably damaged some of the auditory cortices, the TE, and the PER. However, visual cortices were mostly spared. For MEC cannulae, visual cortices were partially damaged. Despite the cortical damage, however, rats performed the tasks very well when aCSF was injected, and this normal performance in the control condition strongly argues against the possibility that MUS effects might be due to the damage in the overlying cortices. Attention and visual capabilities were apparently normal based on the performance data from the control conditions. We also saw no relationships between the amount of damage and the severity of the MUS effects in our study.

## Significance and implications of the current findings

One important caveat of prior behavioral studies was that permanent lesions were produced by the injection of an excitotoxic drug (*Wirth et al., 1998*; *Brun et al., 2008*; *Van Cauter et al., 2013*; *Wilson et al., 2013a*, *Wilson et al., 2013b*) or by the passage of a radio-frequency current (*Kesner and Giles, 1998*; *Parron et al., 2004*; *Van Cauter et al., 2013*) into the LEC or MEC. Permanently damaging a region in the brain by using these methods may induce plastic changes in the functionally related neural circuits, and therefore, the results from these studies may not reflect the normal functions of a brain selectively devoid of the target region (*Ramirez and Stein, 1984*; *Cassel et al., 1997*; *Lavenex et al., 2007*; *Braun et al., 2008*; *Otchy et al., 2015*). The temporary inactivation of a brain region with the use of an inactivating agent, such as MUS or lidocaine, is relatively free from such caveats because normal circuits are preserved during testing. More importantly, the reversible inactivation method allows for the comparison of the normal mode of operation of the circuit (by injecting vehicle solution) with its inactivated mode within the same animal, as opposed to the between-group comparisons that can only be performed in lesion studies. Surprisingly, to our knowledge, no experimental study has used the reversible inactivation for either the LEC or MEC to examine spatial versus nonspatial memory functions in those regions, although some studies have used the methods to test the roles of the LEC in the eye-blink conditioning paradigm (*Morrissey et al., 2012*; *Tanninen et al., 2013*).

To the best of our knowledge, our study is the first to compare the behavioral effects of the reversible inactivation of the LEC and MEC directly within the same animals. The results of the current study largely verified experimental findings separately reported in prior studies and imply that a significant portion of the MEC-dependent deficits previously reported could be attributed to the roles of the MEC in the association of external visual contextual information with egocentric or path-integrative information (*Parron et al., 2004*; *Hafting et al., 2005*; *Hargreaves et al., 2005*; *Brun et al., 2008*; *Van Cauter et al., 2013*). Our findings also indicate that the LEC may play an important role in contextual response to an object using background visual scene or context. This

conjecture may find support in the literature because lesions targeting the LEC did not appear to produce memory deficits when objects and places had to be remembered independently from each other, but produced deficits when objects were required to be recognized in association with places or context (*Deshmukh and Knierim, 2011*; *Hunsaker et al., 2013*; *Tsao et al., 2013*; *Van Cauter et al., 2013*; *Wilson et al., 2013a*, *2013b*).

Our results, together with other studies (*Lavenex and Amaral, 2000*; *Witter et al., 2000*; *Burwell et al., 2004*; *Hargreaves et al., 2005*; *Knierim et al., 2006*; *Kerr et al., 2007*; *Henriksen et al., 2010*; *Ito and Schuman, 2012*; *Hunsaker et al., 2013*; *Tsao et al., 2013*; *Van Cauter et al., 2013*; *Knierim and Neunuebel, 2016*), beg the question of whether the association of the nonspatial and spatial information is the unique role of the hippocampus and suggest that its upstream areas may be engaged in such associative functions as well. The unique contribution of the hippocampus may come from other cognitive processes instead, including pattern completion and separation (*Marr, 1971*; *Lee et al., 2004*; *Leutgeb et al., 2005*), temporal processing of events (*Lee and Wilson, 2002*; *Eichenbaum, 2014*; *Allen et al., 2016*), and value-based decision making (*Lee et al., 2012*; *Mizumori and Tryon, 2015*; *Palombo et al., 2015*; *Schumacher et al., 2016*), when receiving inputs from upstream structures. Our findings also strongly suggest that behavioral response components should be incorporated when theorizing the functions of different regions in the medial temporal lobe. For example, an excellent review by *Knierim et al. (2014)* incorporated path integration and movement information to the spatial information processing stream. Our study suggests that, instead of representing 'what' information, the LEC may process 'what should I do to this object in this context?', whereas the MEC may represent 'where should I go from here in this context?'

The results of the current study, together with our prior studies (*Kim et al., 2012*; *Ahn and Lee, 2014*; *Delcasso et al., 2014*; *Lee et al., 2014*), suggest that visual scene memory tasks provide valuable opportunities for functionally dissociating different areas of the hippocampal memory system. This utility may be important particularly for examining the functions of the EC because prior behavioral studies have taught us that results may be highly variable if a behavioral task allows the animal to choose from multiple strategies (*Johnson and Kesner, 1994*; *Pouzet et al., 1999*; *Burwell et al., 2004*; *Hunsaker et al., 2013*; *Van Cauter et al., 2013*; *Wilson et al., 2013a*, *2013b*; *Rodo et al., 2017*). The scene memory tasks used in the current study do not allow rats to solve the task other than using the surrounding visual scenes in the background and may suffer less from such confounding factors.

## Materials and methods

### Subjects

Seventeen male Long-Evans rats (RRID:RGD_68074) weighing 340–400 g were used. All animals were housed individually and underwent a two-week acclimation period in Plexiglas cages in a temperature- (26°C) and humidity-controlled environment (40–60%). Rats were kept on a 12 hr light-dark cycle. For behavioral experiments, all animals were food-restricted to 80% of their free-feeding weights with free access to water. All protocols (SNU-120925-1-7) were in compliance with the Institutional Animal Care and Use Committee of the Seoul National University.

### Behavioral apparatus

A T-shaped linear track (86 × 40 cm) elevated to 88 cm above the floor was used in a visual scene-based spatial choice (SSC) task in the current study (*Delcasso et al., 2014*). A guillotine door-operated start box (22.5 cm x 16 cm x 31.5 cm) was attached to the end of the stem of the T-track. A food well (2.5 cm in diameter and 0.8 cm in depth) was placed at the end of each arm of the T-track. Three 17-inch LCD monitors were installed as an array to surround the upper portion of the T-track to provide visual background scenes. Four optic fiber sensors (Autonics, Korea) were installed on both side walls of the linear track at different points (1 cm, 27 cm, 47 cm and 67 cm from the start box) to record the animal's position on the track. The activation of the first optic fiber sensor triggered the onset of the scene stimuli on the LCD monitors. The apparatus was surrounded by black curtains, and no external visual cues were provided on the curtains. The experiment room was dimly lit by an overhead array of LED lights (four lx) and two loud speakers played white noise (80 dB)

during behavioral sessions to mask environmental noise. A digital camera installed on the ceiling recorded the behavioral testing sessions and transferred the data to a PC outside the testing room.

The T-track was also used in a scene-based nonspatial choice (SNSC) task. In this task, however, the two arms of the track were made unavailable to the rat by the insertion of a linear track (82.5 cm x 7.5 cm). Two side walls (each 10 cm x 7.5 cm x 0.7 cm) and a food well were found at the end of the track. A wide-mouth amber glass jar (5.8 cm in diameter and 7 cm in height) was placed above the food well. The amber jar contained play sand mixed with cereal powder (Froot Loops, Kellogg's, USA) at a 7:1 ratio.

An object-based nonspatial choice task was conducted in a rectangular acrylic box (55 cm x 23 cm x 23 cm) made of transparent red acrylic panels. A guillotine door-operated start box (22.5 cm x 16 cm x 31.5 cm) was attached to the end of the acrylic box. A linear track (52 cm x 20 cm) was placed inside the box by installing side walls (each 52 cm x 8 cm x 0.5 cm). A food well was placed 10 cm from the end of the linear track. The wide-mouth amber glass jar (filled with play sand and cereal powder) was also used in the task, and a magnet was attached to the glass jar so that an object could be attached to the jar during the task.

## Handling and familiarization

Naïve rats were handled by an experimenter for 30 min a day for 3 days. Following the handling phase, rats were placed on a lab cart (99 cm x 45 cm x 84 cm) to forage for multiple pieces of cereal scattered on the surface of the cart. After acclimation to the environment during the foraging session on the cart for 3 days, rats were familiarized to the experimental room and the behavioral apparatus. During this period, rats were allowed to freely explore the apparatus while consuming cereal pieces scattered over the track and in the food wells. Once rats consumed over 80 pieces of cereal within 30 min for two consecutive days, a shaping phase began.

## Shaping

a. *SSC task*: A rat was first placed in the start box. When the start box door was opened by an experimenter, the rat exited the start box. Once the rat reached the choice point of the T-track, the rat was trained to enter the arm in which a silver metal washer (4 cm in diameter, 0.5 cm in height) had been placed over the food well. Once the rat displaced the washer, a quarter of a piece of cereal was found. The rat was allowed to eat it and was then gently guided back to the start box by the experimenter. The food well in the opposite, unrewarded arm was covered with a black metal washer. Rats were trained until the following criteria were satisfied. First, the median choice latency (from the opening of the start-box door to the displacement of the washer) was less than 5 s. Second, rats could correctly remove the silver washer covering the baited food well in more than 20 out of 40 trials. A daily session was finished when the rat conducted 40 trials or when 30 m had passed, whichever came first.

b. *SNSC task*: A trial began when the experimenter opened the start box. Once exiting the start box, the rat was required to respond by either digging into or pushing the amber jar to obtain a food reward. To shape the digging behavior, rats were allowed to find a piece of cereal placed on the top of the play sand in the jar. The cereal reward was then gradually hidden in the sand over many trials and was eventually completely hidden in the sand (1 cm from the surface of the sand). When the rat found the cereal reward, it was allowed to consume it and was then gently guided back into the start box. For pushing behavior, in the earlier shaping stage, the food well was half-closed by a glass jar so that the rat could see the food reward. The food well was gradually closed by the glass jar over the course of many trials and was eventually completely closed. To shape both digging and pushing responses, rats were trained until the following two criteria were satisfied: first, the median latency from the opening of the start box door to the moment the rat received food reward was less than 5 s. Second, the rat could completely displace the glass jar covering the baited food well (for the pushing response) or could find a food reward completely hidden in the sand in more than 20 out of 40 trials. A daily session was forcibly finished when the rat completed 40 trials or when 30 m passed, whichever came first. Once the criteria were met, pre-surgical training for the main task began.

## Pre-surgical training

a. *SSC task*: A trial began when the rat exited the start box and triggered a sensor placed 1 cm from the door. The activity of the sensor led to a visual scene display (e.g., pebbles or zebra stripes) on the LCD monitors. Rats were trained to choose the right arm for the pebbles scene and the left arm for the zebra stripes scene. A correct choice resulted in a piece of cereal available in the food well (when displaced by the rat). The rat was trained to consume the reward in the start box after being guided back into the start box. A wrong choice resulted in the rat being returned to the start box without a reward. Each scene appeared, in a pseudo-randomly intermixed fashion for 20 trials in a daily session. The rat was trained until its performance level reached greater than 80% correct responses per scene or above 75% correct responses on average for all scenes, with a response bias of less than 0.15. The response bias was calculated by first subtracting the number of right response-associated trials from left response-associated trials and then dividing the difference by the sum of the trials. When the rat reached the performance criterion for the first pair of scene stimuli for two consecutive days, the animal was trained with the second pair of scene stimuli, e.g., mountain and bamboo patterns, until it reached the same criterion level with the first pair of scene stimuli. Then, when the rat reached the performance criteria for both the first and the second scene pairs, all four scenes were pseudo-randomly presented across trials to train the rat until it reached greater than 80% correct responses for each scene.

b. *SNSC task*: Upon opening the start box door and the rat exiting the start box, one of the first pair of scene stimuli, pebbles or zebra stripes, was displayed on the LCD monitors in a pseudo-random sequence. Rats were trained to respond by either digging in the sand or pushing the jar in association with the scene stimulus (e.g., pushing for pebbles and digging for zebra stripes) for 40 trials a day. If the rat stood up and touched the sand with both paws, it was recorded as a digging response. If the rat moved the jar with its snout (even if the food well was not exposed), it was recorded as a pushing response. No correction was allowed when the rat made an error. When the rats reached the performance criterion for the first pair of scene stimuli for two consecutive days, the second pair of scene stimuli (e.g., palm trees and peacock feathers) was introduced for training. The performance criterion for each pair of scene stimuli was greater than 80% correct responses per scene or greater than 75% correct responses for the overall scenes, with a response bias of less than 0.15. When the rats reached the performance criterion for the second pair of scene stimuli, they were trained to the criterion (>80% correct response for each scene) for all four scenes.

## Surgery

Once the rats reached the performance criteria for four scene stimuli for two consecutive days, a 3D-printed cannula complex (20 mm x 9 mm x 26 mm) was surgically implanted. The 3D complex was composed of four stainless steel guide cannulae targeting both the lateral entorhinal cortex (LEC) and medial entorhinal cortex (MEC) bilaterally. First, the rat was deeply anesthetized with an intraperitoneal injection of sodium pentobarbital (Nembutal, 65 mg/kg), and its head was fixed in a stereotaxic frame (Kopf Instruments, USA). Anesthesia was then maintained by isoflurane inhalation (0.5–2% isoflurane mixed with 100% $O_2$) throughout the surgery. An incision was made along the midline of the scalp and the skull was exposed. The skull surface was leveled after adjusting the levels of bregma and lambda on the same horizontal plane. Six small burr holes were made on the skull to place anchoring skull screws to hold the 3D-printed cannula complex with dental cement. Four burr holes were also drilled to insert the guide cannulae of the cannula complex into the LEC and the MEC at the following coordinates: 5.9 mm posterior to bregma, 6.8 mm lateral to midline, 6.4 mm ventral from dura for the LEC, and 7.8 mm posterior to bregma, 4.8 mm lateral to midline, 5.9 mm ventral from dura. Following insertion of the guide cannulae into the target areas, medical grade silicone (Kwik-Sil, World Precision Instruments, USA) was applied to fill the burr holes to block the influx of dental cement into the exposed area of the brain during the cementing procedures. Dental cement was then applied to the skull surface to firmly hold the 3D-printed cannula complex. The rat was allowed to recover for 7 days before behavioral testing began.

## Post-surgical testing and drug injection

For both SSC (n = 8) and SNSC (n = 9) tasks, after recovery from surgery, rats were retrained to the pre-surgical criteria with the four scene stimuli for two consecutive days. Then, on the next day,

artificial cerebrospinal fluid (aCSF) was injected into the EC (i.e., both the LEC and the MEC, 0.3 μL per site) 20 m before behavioral testing. The vehicle solution was injected into both the LEC and MEC in the current study because our prior pilot studies showed no difference in performance between the two conditions. More importantly, prolonging the testing period with aCSF might carry the risk of overtraining the rats, which we wanted to avoid to properly test the functions of the LEC and MEC while those areas were engaged in the normal performance of the task. Over the next two days, the same animal received an injection of the GABA-A receptor agonist muscimol (MUS) into the LEC and then into the MEC (0.3 μL per site), or vice versa (the injection sequence was counterbalanced among the rats).

## Histology

When all experimental testing was completed, fluorophore-conjugated muscimol (f-MUS; Sigma, USA) was injected bilaterally into both the LEC and MEC (0.3 μL per site) to verify the diffusion range of the inactivating agent. Twenty minutes after the injection of f-MUS, rats were killed by $CO_2$ overdose and were perfused transcardially with phosphate-buffered saline, followed by a 4% v/v formaldehyde solution. The brain was extracted from the skull and was soaked in 4% v/v formaldehyde-30% sucrose solution at 4°C until it completely sank. The brain was then gelatin-coated and soaked again in a 4% v/v formaldehyde solution-30% sucrose solution at 4°C until it completely sank. Then, the brain was cut into 40 mm coronal sections on a freezing microtome (HM 430, Thermo Fisher Scientific, USA). Every second section was collected for thionin staining to verify the position of the 3D complex cannula, and every third section was used to examine the diffusion of f-MUS with a fluorescent microscope (Eclipse 80i, Nikon, Japan).

## Data analysis

Performance was quantified as the proportion of correct trials in each session. Two-way repeated-measures ANOVA was used to examine the effects of the drugs. A t-test corrected for multiple comparisons was used to examine the presence of any significant differences according to injection sequences. The response latency was defined as the duration from the time that the animal exited the start box to the time when the animal began to exhibit behavioral responses to obtain food rewards. The proportions of correct trials over the overall trials were regarded as behavioral performance assessments for each session.

## Acknowledgements

This study was supported by the Brain Research Program (NRF-2015M3C7A1031969), the Basic Science Research Program (NRF-2013R1A1A2062882 and NRF-2016R1A2B4008692), the Science Research Center grant (2014051826), the BK21+ program through the National Research Foundation of Korea. We thank Choong-Hee Lee for his statistical help and Eun-Hye Park for her helpful comments for the manuscript.

## Additional information

### Funding

| Funder | Grant reference number | Author |
| --- | --- | --- |
| National Research Foundation of Korea | 2013R1A1A2062882 | Inah Lee |
| National Research Foundation of Korea | 2016R1A2B4008692 | Inah Lee |
| National Research Foundation of Korea | 2014051826 | Inah Lee |
| National Research Foundation of Korea | BK21 plus program | Inah Lee |
| National Research Foundation of Korea | 2015M3C7A1031969 | Inah Lee |

The funders had no role in study design, data collection and interpretation, or the decision to submit the work for publication.

## Author contributions

S-WY, Data curation, Formal analysis, Visualization, Methodology, Writing—original draft, Project administration; IL, Conceptualization, Resources, Data curation, Formal analysis, Supervision, Funding acquisition, Validation, Investigation, Visualization, Methodology, Writing—original draft, Project administration, Writing—review and editing

## Author ORCIDs

Inah Lee, http://orcid.org/0000-0003-3760-4257

## Ethics

Animal experimentation: This study was performed in strict accordance with the guidelines of the Institutional Animal Care and Use Committee of the Seoul National University. All protocols (SNU-120925-1-7) were in compliance with the Institutional Animal Care and Use Committee of the Seoul National University.

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
