## [Decision Letter]

Thank you for submitting your article "Functional double dissociation within the entorhinal cortex for visual scene-dependent choice behavior" for consideration by *eLife*. Your article has been reviewed by two peer reviewers, and the evaluation has been overseen by a Reviewing Editor and Sabine Kastner as the Senior Editor. The following individual involved in review of your submission has agreed to reveal his identity: Alex Easton (Reviewer #2).

The reviewers have discussed the reviews with one another and the Reviewing Editor has drafted this decision to help you prepare a revised submission.

The referees find this an interesting inactivation study in the topical area of differentiating the functions of lateral and medial Entorhinal cortex (LEC and MEC). However, both have some significant issues that need to be addressed (see comments copied below).

One concerns the power of the results, specifically the weakness of the LEC inactivation effect: the fact that it is only present in 3 out of 6 animals, and might potentially be due to spread of muscimol to area X, where area X is closer to LEC than to MEC (and influences digging/ pushing) in some animals. The issue being that the much smaller non-flourescent muscimol molecules would diffuse further than shown by the f-muscimol. Is a solution to perform the behavioral study using f-muscimol in a few more animals? Referee 3 requests a diagram showing implantation points for all animals, for comparison with the supplemental data on performance regarding this point.

The other concerns the interpretation of the findings – what is the critical task difference: is digging (down) or pushing (ahead) really so different than turning left or right? Is this difference due to being more/less spatial? Are there any videos or descriptions of the way in which the LEC animals fail the 'non-spatial' scene task, so that we could try to understand why they fail?

Main comments (Referee 2):

This is a fascinating result from a careful piece of work and is of very high interest to a wide range of researchers. However, the implications rely heavily on the detail of the methods, which can be sparse at times. It is important that the authors clarify several things to allow the importance of the findings to be fully understood:

1) Numbers of animals in each task are small – is there sufficient power in these results? Some form of power analysis is important. If one looks at the data for (for example) the SNSC task in Figure 1, the error bars suggest that there is reasonable overlap in the performance of the groups and whilst the means are statistically different from one another, without understanding the power of the study it is possible with such low numbers that this would be a 'fluke'. Whilst the double dissociation makes this somewhat unlikely it does need to be addressed.

2) To what extent are the object and scene tasks comparable? Much of the interpretation relies on the object based version of the tasks being unaffected by temporary inactivation of either area – suggesting the effect is limited to scenes. However, power remains to be addressed (point 1) but also performance is much higher in the object task. One assumes it is easier given the stimuli are likely more discriminable. Therefore, is it that a dissociation exists between LEC and MEC in this task, but is merely hidden by ceiling effects in performance?

3) There is significant damage dorsal to the infusion sites from the cannula (one assumes). Although unlikely to have a significant bearing (as the animals *can* do the task when muscimol is not infused) the authors do need to consider any relevance of this extraneous damage – especially as it may be damaging attention and/or visual systems which will impact on normal task performance.

Main comments (Referee 3):

Using a within-subject design in which the same rats received local intra-cerebral injections of muscimol, the authors assessed the functional role of the medial entorhinal cortex (MEC) and lateral entorhinal cortex (LEC) in two tasks that both relied on the identification of visual scenes but differed in the type of response that was required. In the first (spatial) task, the visual scene told the rat whether to turn right or left in a T-maze. In the second task, the visual scene told the rat whether to push a jar or to dig in that jar. The results revealed that MEC inactivations induced a strong deficit in the first task, but not in the second task and that LEC inactivations induced a relatively smaller deficit in the second task, but no deficit in the first task. The authors' conclusions question standard hypotheses about MEC and LEC functions based on their anatomical connections and on previous behavioral data.

This is an interesting piece of work that will usefully complement the growing body of literature tackling the functional roles of the MEC and LEC. My main concern is that it does not however provide any insight into what these functions might be. While the results of the first task (scene-based spatial choice task) and clear and fit existing data, those of the second task (scene-based non-spatial choice task) are more difficult to interpret.

First, it is not clear why the authors designed this task (besides the idea of having a non-spatial task), what their prediction was, and what the specific function that seems to be handled by the LEC might be. The discussion does not provide any hint as to this last point.

Second, the data are a bit weak as the deficit induced by LEC inactivations is relatively small (as noticed by the authors) and does not appear to be very robust. Looking at the individual data of the six rats that were tested revealed virtually no deficit in 3 of the rats.

In addition to these main concerns, I would like to see a diagram showing the implantation points for all rats. Although Figure 1 is useful, it does not discard the possibility that some of the reported effects are due to diffusion of muscimol into neighboring structures (including MEC). The visualization of diffusion extent is useful but not sufficient as the molecular weight of fluorescent muscimol (ca 600) is much greater than that of regular muscimol (ca 100), leading to the possibility of underestimation of diffusion range.

---

## [Author Response]

*The referees find this an interesting inactivation study in the topical area of differentiating the functions of lateral and medial Entorhinal cortex (LEC and MEC). However, both have some significant issues that need to be addressed (see comments copied below).*

*One concerns the power of the results, specifically the weakness of the LEC inactivation effect: the fact that it is only present in 3 out of 6 animals, and might potentially be due to spread of muscimol to area X, where area X is closer to LEC than to MEC (and influences digging/ pushing) in some animals. The issue being that the much smaller non-flourescent muscimol molecules would diffuse further than shown by the f-muscimol. Is a solution to perform the behavioral study using f-muscimol in a few more animals?*

Regarding the power of the results:

To address this concern of the Reviewing Editor (also raised by both reviewer 2 and reviewer 3), we ran four more rats in the SNSC task during the revision period of the manuscript. One of the rats was not usable because the rat’s cannula in the left LEC was blocked by a broken dummy cannula, but three rats’ data were successfully obtained and their cannula positions were located in the LEC correctly (Rats# 455, 457, and 458 in our raw data summary table). Adding these rats’ data to the original data set made the statistical results more significant compared to our original manuscript. That is, a repeated-measures ANOVA showed highly significant effects of the drug condition (F_(2,16)_=10.1, p = 0.001; improved from p = 0.02 in the original manuscript) and there were significant differences in performance between aCSF and LEC-MUS conditions (p < 0.001; improved from p < 0.01 in the original manuscript) and also between LEC-MUS and MEC-MUS conditions (p < 0.01; improved from p = 0.05 in the original manuscript). We revised our manuscript accordingly (subsection “Double dissociation between the LEC and MEC in visual scene-based choice behavior”, first paragraph).

Although the additional data gave us more confidence regarding the roles of the LEC in the SNSC task, we observed the same pattern of results that we saw during the original experiment in the rats additionally run during the revision period. Specifically, two of the rats (rat# 455, 457) in these rats showed severe performance deficits, but one rat (rat# 458) showed only a mild deficit in the SNSC task although all three rats had their LEC cannulae implanted in the LEC. We observed this pattern repeatedly, where some rats were severely impaired while other rats were only mildly affected by MUS in the LEC in both the original experimental period and the additional data-collecting period. The large individual difference underlies the greatest variability in performance data in the LEC-MUS condition in the current study (revised Figure 3). Therefore, we checked whether cannula positions in different rats might have affected the amount of severity in performance deficit. Figure 1 of our revised manuscript shows the LEC cannula tip positions in all the rats (n=9). Comparing the cannula positions together with the summary table for the raw data suggests that the rats (rat# 362,404,417,458) showing seemingly mild or no deficits in the LEC-MUS condition compared to the aCSF condition had their LEC cannulae implanted mostly in the lateral edges of the LEC along the superficial layers. However, the LEC cannula-tips of the rats (rat# 341,361,403,455,457) that showed relatively severe deficits with LEC-MUS compared to the aCSF condition were located more medially targeting deeper layers in the LEC (revised Figure 1). Confirming these observations, when we compared the performance levels of the rats with medially vs. laterally placed LEC cannulae, there was no significant difference in performance between the two groups in aCSF conditions (t_(7)_ = 0.3, p = 0.77) (revised Figure 3). However, MUS injections into the LEC induced a significant difference in performance between the medial and lateral cannula groups (t_(7)_ = -4.35, p < 0.01) (revised Figure 3). We added these newly discovered important findings in our revised manuscript (subsection “Double dissociation between the LEC and MEC in visual scene-based choice behavior”, third paragraph).

The results may suggest that the LEC deficits in the SNSC task could be layer-specific or that MUS injected through superficially implanted LEC cannulae might have been leaked to the brain surface. These possibilities are discussed in our revised manuscript (subsection “Visual scene-based disambiguation of object-associated responses in the LEC”, second paragraph).

Regarding the drug diffusion issues:

The Reviewing Editor expressed his/her concern that the performance deficits in LEC-MUS might be spurious (because only half of rats showed deficits in the original manuscript) and suggested that this could be due to the uncontrolled spread of MUS into an unknown area closer to the LEC in those rats showing performance deficits. First of all, in our response above, we verified the originally reported results by showing that adding more rats (n=3) to the original data set made our statistical results more compelling compared to the original manuscript (with p-values improved from 0.02 to 0.001 in repeated-measures ANOVA). Second, we also showed in our revised manuscript that the rats showing seemingly intact performance in the LEC-MUS condition had their cannulae mostly in the lateral edges of the LEC along more superficial layers. In contrast, the rats showing more severe performance deficits in the same drug conditions had their cannula tips placed more medially along deeper layers of the LEC. Based on our histological results, we think it is unlikely that our LEC-MUS impaired performance through other unknown areas because LEC cannula tips were all within the LEC region. Third, regarding the comments of R.E. that the spread of fluorescent MUS (f-MUS) might be underestimating the real diffusion range of MUS, we completely agree with the comment, and this has been shown by the Brown group (Allen et al., 2008). However, for the suggestion of running additional rats with f-MUS injected in the LEC, our prior experience with prolonged injections of f-MUS in rats in this type of behavioral paradigm (i.e., injecting MUS and testing rats for multiple days) taught us that such protocol sometimes caused health problems in rats for unknown reasons. In that case, it is difficult to attribute resulting performance deficits specifically to inactivation of the local area. Furthermore, injecting f-MUS for behavioral testing requires usually 40 min of diffusion time (which is twice longer than the MUS injection protocol), compared to 20 min with regular MUS in the current study, and the lipophilic nature of f-MUS makes drug spread uneven (Allen et al., 2008). Our experience is also that f-MUS appears to show bigger upward spread along the cannula track compared to regular MUS and thus may require larger injection quantity compared to regular MUS. With these known caveats of using f-MUS in behavioral testing, we decided not to run our rats with f-MUS injections. Since the fluorescent photomicrographs might underrepresent the real diffusion range of MUS (and this is visible in our histological data) and the cannula tracks are clearly visible in the photomicrographs of the Nissl-stained sections, we removed the fluorescent photomicrographs in our revised manuscript.

In sum, based on our (i) additional behavioral data showing highly significant, replicable effects of LEC-MUS, (ii) additionally reported relationships between LEC cannula tip positions and performance levels, (iii) highly significant double dissociation between LEC and MEC within the same rats (revised Figure 3) (even though the two areas are adjacent to each other), we are confident that the performance deficits in the SNSC task with LEC-MUS were genuine.

*Referee 3 requests a diagram showing implantation points for all animals, for comparison with the supplemental data on performance regarding this point.*

Please see our detailed responses to the comments of the Reviewing Editor above and updated contents of our revised manuscript.

*The other concerns the interpretation of the findings – what is the critical task difference: is digging (down) or pushing (ahead) really so different than turning left or right? Is this difference due to being more/less spatial? Are there any videos or descriptions of the way in which the LEC animals fail the 'non-spatial' scene task, so that we could try to understand why they fail?*

We included video clips showing the normal trials under aCSF and error trials when MUS injected in LEC (Figure 3; Video 2). As can be seen in the video clip, one big difference that is obvious when one compares the response types required between the SSC task (left vs. right turns required for choice; Figure 2; Video 1) and SNSC task (digging vs. pushing required to an object, the sand-filled jar, for choice) is that the animal’s response is registered through the object in the SNSC task, but not in the SSC task. That is, in the SSC task, the rat’s response is purely spatial, but not in the SNSC task. Once trained, the rat does not stop at the choice point of the T-maze before making a choice in the SSC task, but it does pause briefly in front of the sand-filled jar before making a response in the SNSC task. When one observes the error trials in the SNSC task in the video (starting from 43s in the video), compared to the normal trials, the rat injected with MUS in the LEC appeared to make hesitating or up/down responses in front of the jar. In the zebra-stripe trial (starting from 53s in the video), the rat even correctly chose to dig in the sand initially but then came down and push the jar to make an error (which suggests that the rats were not using olfactory cues from the reward buried in the sand in our task). This behavior is observed when the rats were injected with MUS in the LEC and may suggest that LEC inactivation may play critical roles when an animal chooses a response to an object in a contextual manner. Our study suggests that the digging-pushing behavior itself is not LEC-dependent because the LEC was not required when the same behavioral responses were cued by an object, but not visual context, in the object-based choice task (Figure 4).

*Main comments (Referee 2):*

*This is a fascinating result from a careful piece of work and is of very high interest to a wide range of researchers. However, the implications rely heavily on the detail of the methods, which can be sparse at times. It is important that the authors clarify several things to allow the importance of the findings to be fully understood:*

*1) Numbers of animals in each task are small – is there sufficient power in these results? Some form of power analysis is important. If one looks at the data for (for example) the SNSC task in Figure 1, the error bars suggest that there is reasonable overlap in the performance of the groups and whilst the means are statistically different from one another, without understanding the power of the study it is possible with such low numbers that this would be a 'fluke'. Whilst the double dissociation makes this somewhat unlikely it does need to be addressed.*

Most of the comments of reviewer 2 have been addressed while responding to the comments of the Reviewing Editor as described above. Reviewer 2’s comment on the seemingly overlapping error bars of performance data in our original Figure 2 (Reviewer 2 mentioned Figure 1 but we believe he/she meant Figure 2) turned out to be an important one, and we redrew all of our performance data using box plots instead of bar graphs (revised Figure 3). The box plot of the LEC-MUS condition made the large variance in the data more visible, and this turned out to be attributable to the fact that performance levels of some rats were severely affected by LEC-MUS than those of other rats. This discrepancy was related to their LEC cannula positions based on our additional analysis (Figure 1 and Figure 3). See details in our response to the Reviewing Editor above. Power analysis might not tell us much information regarding this issue mainly because our study used a within-subject testing paradigm. Specifically, based on our power analysis with the following parameters (effect size f=5.505; α error probability=0.05; power=0.95; the *number of group=1*; the number of measurements=3; correlation among representative measures=0.360; non-sphericity correction=1), one could obtain the actual power of 0.99 with only the sample size of 2. Our extensive experience with behaviorally testing rats with within-subject testing designs has taught us that n of 6 to 8 is usually sufficient to draw firm conclusion when using behavioral tasks in which rats were pre-trained to criterion across many days before being tested with drug injections (Lee & Solivan, 2008; Jo & Lee, 2010; Kim & Lee, 2011; Yoon et al., 2011; Kim et al., 2012; Lee & Shin, 2012; Lee et al., 2014). This is in contrast to those behavioral testing paradigms that require no or minimal pre-training such as fear conditioning because individual variability is often bigger in those one-shot paradigms compared to our paradigm.

*2) To what extent are the object and scene tasks comparable? Much of the interpretation relies on the object based version of the tasks being unaffected by temporary inactivation of either area – suggesting the effect is limited to scenes. However, power remains to be addressed (point 1) but also performance is much higher in the object task. One assumes it is easier given the stimuli are likely more discriminable. Therefore, is it that a dissociation exists between LEC and MEC in this task, but is merely hidden by ceiling effects in performance?*

Power issues have been addressed above (see our responses to reviewers in (2)). Regarding reviewer 2’s comment on the possibility that the object memory task might be easier than the scene-based tasks, we respectfully disagree with the reviewer’s comment. First of all, reviewer 2 mentioned that performance was much higher in the object task, but we do not understand what this comment was based on. Specifically, when comparing aCSF performance across the tasks, we see no significant differences among the tasks; the mean performance was 92.8% for SSC, 91.7% for SNSC, and 92.9% for the object task (F_(2,23)_=0.13, p = 0.88; ANOVA). So, normal performance was not much higher in the object task compared to the SSC and SNSC tasks. More importantly, in our recent study using the same object task for testing the differential roles of the perirhinal cortex and postrhinal cortex, there was dissociation in performance under MUS infusions in the object memory task (Park et al., 2016, SFN); MUS significantly impaired performance when injected into the perirhinal cortex, but not in the postrhinal cortex. These results may argue against the comment of the reviewer (i.e., ceiling effect-based null effect), and we have added these contents in our revised manuscript (subsection 2 No involvement of the LEC and MEC when an object alone cued choice behavior”, last paragraph).

*3) There is significant damage dorsal to the infusion sites from the cannula (one assumes). Although unlikely to have a significant bearing (as the animals can do the task when muscimol is not infused) the authors do need to consider any relevance of this extraneous damage – especially as it may be damaging attention and/or visual systems which will impact on normal task performance.*

We agree with the reviewer that there is a concern for this in all cannula-implantation experiments. Since the cannulae should penetrate the overlying cortices during the implantation surgery, it is inevitable to cause some damage over the cortical areas as R2 mentioned. Specifically, based on our surgical coordinates, the LEC cannulae inevitably damaged some of the auditory cortices, the TE, and the perirhinal cortex. Visual cortices were mostly spared. For MEC cannulae, some visual cortices were damaged. Importantly, despite the cortical damage, rats performed the tasks very well when aCSF was injected, and this normal performance in the control condition strongly argues against the possibility that MUS effects were due to the overlying cortical damage. Attention and visual capabilities were obviously normal based on the performance data from the control conditions. We also see no relationships between the amount of cortical damage and the severity of the MUS effects in our study. We think mentioning this is important in our manuscript and added the related discussion (subsection 2 Visual scene-based disambiguation of object-associated responses in the LEC”, last paragraph).

*Main comments (Referee 3):*

*[…] This is an interesting piece of work that will usefully complement the growing body of literature tackling the functional roles of the MEC and LEC. My main concern is that it does not however provide any insight into what these functions might be. While the results of the first task (scene-based spatial choice task) and clear and fit existing data, those of the second task (scene-based non-spatial choice task) are more difficult to interpret.*

*First, it is not clear why the authors designed this task (besides the idea of having a non-spatial task), what their prediction was, and what the specific function that seems to be handled by the LEC might be. The discussion does not provide any hint as to this last point.*

We agree with the comments that our initial manuscript did not explain the rationale and implications of the study well. We have extensively revised our manuscript, and the current version explains the above points better. Briefly, our motivation behind the study was initially to test whether the LEC and MEC would be differentially involved in a visual scene memory task. Since our lab has shown that the dorsal hippocampus is critical for using a visual scene to make a spatial or nonspatial choice conditionally, we wanted to test whether its upstream input regions also played critical roles. We’d like to emphasize that visual scene has never been systematically used in rodent literature for testing the entorhinal functions and our study is the first one that systematically used the visual scene as the major variable for testing the differential functions of the LEC and MEC. Prior studies assumed that visual scene could be used during spatial navigation, but this has never been directly tested.

We observed the MEC-dependent performance deficits, but no significant deficit was found with the LEC-MUS in the SSC task. Since this could be (a) because the MEC, but not the LEC, is critical for scene information processing or (b) because the MEC, but not the LEC, is critical for making spatial responses, or both. We originally tested the rats in a tactile-version of the T-maze task and confirmed that the rats with the MEC inactivated with MUS were normal compared to the aCSF condition in making spatial choices in the T-maze by using a tactile cue (wire mesh or sandpaper in the stem of the maze). Considering the reviewer’s comment, these results are important to show that the MEC’s involvement is specific to the condition where visual scenes should be associated with spatial responses and we added the results from the tactile memory task in our revised manuscript (Figure 2 and subsection “Double dissociation between the LEC and MEC in visual scene-based choice behavior”, first paragraph). Then, our question became whether the MEC was also important when visual scene should be associated with nonspatial choice behavior. So we tested rats in a different scene-based task (i.e., SNSC task) that had relatively little spatial response component. We used the SNSC task specifically because we knew through our previous study that inactivating the dorsal hippocampus prevented rats from exhibiting normal performance in that task. As a result, we saw that the LEC-MUS impaired performance in the SNSC task, whereas no impairment was seen with the MEC-MUS in the same rats.

We revised our manuscript extensively to include this rationale and theoretical implications of our findings: see changes in our Abstract, Introduction, “Significance and implications of the current findings” in the Discussion of our revised manuscript. The following excerpt from our Discussion may help the reviewer to understand our viewpoint better:

“Our findings also strongly suggest that behavioral response components should be incorporated when theorizing the functions of different regions in the medial temporal lobe. […] Our study suggests that, instead of representing “what” information, the LEC may process “what should I do to this object in this context?”, whereas the MEC may represent “where should I go from here in this context?””

*Second, the data are a bit weak as the deficit induced by LEC inactivations is relatively small (as noticed by the authors) and does not appear to be very robust. Looking at the individual data of the six rats that were tested revealed virtually no deficit in 3 of the rats.*

*In addition to these main concerns, I would like to see a diagram showing the implantation points for all rats. Although Figure 1 is useful, it does not discard the possibility that some of the reported effects are due to diffusion of muscimol into neighboring structures (including MEC). The visualization of diffusion extent is useful but not sufficient as the molecular weight of fluorescent muscimol (ca 600) is much greater than that of regular muscimol (ca 100), leading to the possibility of underestimation of diffusion range.*

Please see our detailed responses to the comments of the Reviewing Editor above and updated contents of our revised manuscript.